# A Longitudinal Study of a Multicomponent Exercise Intervention with Remote Guidance among Breast Cancer Patients

**DOI:** 10.3390/ijerph17103425

**Published:** 2020-05-14

**Authors:** Xiaosheng Dong, Xiangren Yi, Meng Ding, Zan Gao, Daniel J. McDonough, Nuo Yi, Wenzhen Qiao

**Affiliations:** 1Department of Sport and Health, School of Physical Education, Shandong University, Jinan 250061, China; dxiaosheng@hotmail.com (X.D.); xrenyi@sdu.edu.cn (X.Y.); 2College of Physical Education, Shandong Normal University, Jinan 250014, China; dingmeng@sdnu.edu.cn; 3School of Kinesiology, University of Minnesota, Minneapolis, MN 55455, USA; gaoz@umn.edu (Z.G.); mcdo0785@umn.edu (D.J.M.); 4Department of Kinesiology, College of Health Science University of Wisconsin, Milwaukee, WI 53201, USA; nuoyi@uwm.edu; 5Department of Science and Technology, Shandong Institute of Commerce and Technology, Jinan 250103, China

**Keywords:** breast cancer, cardiorespiratory endurance, muscle strength, physical activity, quality of life

## Abstract

*Purpose*: Breast cancer patients in treatment suffer from long-term side effects that seriously influence their physical and mental health. The aim of this study was to examine effectiveness of a 12-week multicomponent exercise (ME) with remote guidance intervention on health-related outcomes after one year among breast cancer patients. *Methods*: In phases I–III, 60 patients (51.2 ± 7.9 years) with breast cancer (BC) who completed chemotherapy/postoperative radiotherapy within the previous four months to two years were randomly assigned to (1) multicomponent exercise with remote guidance (ME) and (2) usual care (UC). Eligible participants were approached to assess cancer-related quality of life (QOL), muscle strength, cardiorespiratory endurance, and physical activity (PA) barriers after one year. *Results*: The results demonstrated that, after one year, the ME group reported higher vitality-related QOL (5.776, 95% confidence interval [CI] 0.987, 10.565, effect size [ES] = 0.360), mental health-related QOL (9.938, 95% CI 4.146, 15.729, ES = 0.512), leg strength and endurance (2.880, CI 1.337, 4.423, ES = 0.557), and strength and endurance of upper extremities (2.745, 95% CI 1.076, 4.415, ES = 0.491) and lower physical activity (PA) hindrance (5.120, 95% CI 1.976, 8.264, ES = 0.486) than the UC group. *Conclusions*: The ME group observed significant differences from the UC group in QOL, muscle strength, cardiopulmonary endurance, and PA participation. These findings suggested that the multicomponent exercise intervention with remote guidance produced long-term health benefits for breast cancer patients.

## 1. Introduction

Breast cancer ranks as the fifth leading cause of cancer mortality among women around the world [1,2]. With advances in screening, diagnosis, and treatment of breast cancer, survival rates for breast cancer survivors have improved and the population of survivors has increased dramatically [3]. However, the occurrence of breast cancer is still increasing, and postoperative patients experience long-term side effects including recurrence, fatigue, lymphoedema, and a decline in physical function, which all seriously affect their physical and mental health [4,5]. Therefore, it is imperative to adopt appropriate treatment to better facilitate disease management in this population [6].

As known, it is a beneficial treatment approach for breast cancer patients to adopt mobile social media and remote videos to conduct exercise guidance that can improve patients’ muscle strength, cardiopulmonary endurance, mood, and quality of life (QOL) [7,8,9]. Some studies have illustrated that exercise improves the short-term welfare of breast cancer patients, but little is known regarding its long-term effects. Patients with breast cancer often suffer from long-term psychological and physiological issues and complications after operations [10,11,12,13]. Therefore, the longitudinal effects of internet-based exercise experiments are of great clinical significance among breast cancer patients.

Previous studies have shown long-term benefits of physical activity in breast cancer patients, providing important information for the treatment of tumors [14,15,16]. Currently, aerobic exercise is mainly recommended for improving cardiovascular function, but it cannot train upper body muscle strength. Because of considering the limited function of lymphedema and upper limb in patients with breast cancer after surgery, the combination of aerobic and resistance exercise is a better beneficial strategy for breast cancer health care [17]. However, these exercise interventions have not utilized any social media and/or remote guidance. Compared with on-site rehabilitation, distance guidance for sports rehabilitation is more convenient without considering the problems such as site and transportation and reduces the economic burden of patients. The previous development level of the Internet cannot realize distance guidance, and it has become possible now with the popularization and wide application of the Internet. It has been found by our previous study that the application of internet-based combination exercises had a short-term benefit in postoperative patients with breast cancer after a 12-week intervention treatment [18]. However, there are few studies with follow-up after intervention because it is very difficult to get in touch with patients. We did a long-term follow-up after the remote exercise guidance intervention and found whether breast cancer patients who underwent distance exercise intervention had a long-term benefit.

In this study, a long-term follow-up was used for comparison of remotely guided multiple exercise combinations (i.e., anti-resistance training, aerobic exercises, and psychological guidance) and usual care, aiming to identify the long-term effects of remotely guided exercise combinations on postoperative patients with breast cancer and to deliver new perceptions of rehabilitation methods and approaches for breast cancer patients. The purpose of this study was to identify whether there were significant differences between the remotely-guided multicomponent exercise (ME) intervention and usual care (UC) condition in muscle strength, cardiorespiratory endurance, QOL, and physical activity (PA) barriers after one-year postintervention follow-up.

## 2. Materials and Methods

### 2.1. Participants

The present study was a longitudinal study employing a one-year follow-up period after 12-week intervention. Participants were randomly chosen and recruited from the Department of Breast Surgery at the Second Hospital of Shandong University, Jinan, Shandong Province from August 2017 to February 2018. Participants were randomized to ME or UC. Eligibility criteria were as follows: women with breast cancer (*n* = 60, age = 51.2 ± 7.9 years) within stages I–III who have completed chemotherapy/postoperative radiotherapy within the previous four months to two years. Conversely, patient exclusion criteria were as follows: communication or language barriers, incomplete questionnaire, metastasis, mental disorder, record of acute suicidality, cognitive brain organic lesion and dementia, inability to use smartphone apps or tele-video, and special physical activity training except for intervention program. Patients with one or more of the above criteria were excluded from this study. All trials were operated in accordance with ethical standards and were approved by the Ethics Committee of the Second Hospital of Shandong University (registration numbers: KYLL-2017(KJ) P-0003). The trial was registered with Chinese Clinical Trial Registry (ChiCTR-IPR-17012368). All individual participants signed an informed consent form in the study. After a 12-week intervention, eligible participants were recalled for evaluation to complete an in-clinic physical fitness test and an online self-reported questionnaire.

### 2.2. Designed Intervention Program

The randomized controlled trial (RCT) protocol [19] and health outcomes of the 12-week exercise intervention [16] have been published elsewhere. Further, the 12-week internet-based multicomponent exercise intervention has been reported previously [18]. Remote exercise rehabilitation guidance included resistance training, cardiorespiratory endurance training, and rehabilitation knowledge. The intervention program in the ME group consisted of the following: (1) resistance training, including muscle strength, endurance, and function training: each session was 30 min, three times per week, including a five-min warm up, 20 min of training, and five-minutes of relaxation. Participants were asked to do 8–12 repetitions at an intensity of 70–80% of their estimated one repetition maximum (1-RM); (2) cardiorespiratory endurance training, including remote aerobic exercise, performed four times per week and measured by rate of perceived exertion (RPE 13–16); and (3) rehabilitation knowledge, special exercise, and health knowledge for BC rehabilitation transmitted by social media apps each day in order to encourage consistency of exercise. Patients were not allowed to participate in any other type of exercises or sports physical activity training program except for the intervention program. The intervention rehabilitation program was directed by professional physiotherapists within the hospital. Various symptoms and performance of patients were recorded and reported to the research coordinator, who determined whether further measures were necessary for patients.

Patients in the UC group were administered general treatment and rehabilitation in terms of treatment requirements. We did not restrict their voluntary movement or general physical activity. The treatment and rehabilitation were directed from the National Institute for Health and Care Excellence Clinical Guidance (NG101) [20].

### 2.3. Follow-Up Period

After completion of the 12-week intervention program, exercise prescriptions consisting of aerobic and resistance exercises were provided to participants in the ME group and simple PA equipment (e.g., elastic bands and dumbbells) was also provided, free of charge. The exercise and health knowledge of BC rehabilitation were delivered by social media apps during the 1-year follow-up period (2018 to 2019).

### 2.4. Outcome Measures

Patient characteristics were reported at baseline, including clinical records and demographic characteristics: age, hemodynamics, blood pressure (blood pressure systolic [BPS] and blood pressure diastolic [BPD]), stage of illness, phase of treatment (observation, chemotherapy, radiation therapy, and radio-chemotherapy). Outcomes were measured at baseline (0 week), postintervention (12 weeks postbaseline), and one-year follow-up. Cancer-related QOL was assessed using the Short Form Health Survey (SF-36) [21,22]. Upper and lower body strength were evaluated by the curl-up test or arm-lifting test (30 s dumbbell of 5 pounds or 2.3 kg lifting test) and the chair stand test, respectively [19]. Cardiorespiratory endurance was measured by maximal oxygen uptake, which was evaluated by the modified Bruce treadmill protocol [23]. The participants’ PA estimates were assessed by a standard self-report questionnaire [24].

### 2.5. Statistical Analysis

The original power calculation was based on our preliminary trial. The vitality score was the primary outcome measure in the 12-week intervention. According to the previous study, we calculated that thirty participants were enrolled in each group for an 80% power to detect a difference in vitality of 8 between the groups with standard deviation (SD) of 10, α level of 0.05, and allowing for a 25% loss to follow-up. Quality of life, PA estimates, muscle strength and cardiopulmonary endurance at 12 weeks of data are available at: www.mdpi.com/xxx/s1 (see Appendix A).

Baseline characteristics were analyzed with analysis of variance (for continuous measures) and the chi-square test (for nominal data) to assess for differences among the two groups. Analysis of linear mixed models was performed for the comparison of health outcomes between groups. Mean differences and 95% confidence intervals (CIs) were accompanied by standardized effect size (ES) (*p* < 0.05 represents significant difference). Effect sizes (Cohen’s d) were categorized as small if *d* > 0.2, as medium if *d* > 0.5, and as large if *d* > 0.8 [25]. Data analysis was performed using SPSS software version 25.0 (IBM Corp., Armonk, NY, USA). All significances were set at 0.05 in this study.

## 3. Results

### 3.1. Participant Characteristics

The flow chart for the intervention process is illustrated (Figure 1). During the period of the 1-year follow-up, seven participants withdrew from the intervention group and nine participants withdrew from the control group. The total dropout rate was 23% in the intervention group and 30% in the control group. The main reasons for dropouts in the intervention group were quitting exercise (*n* = 1), health issues (*n* = 4), or too busy (*n* = 2), whereas the terminated participants in the control group were too busy (*n* = 2), health issues (*n* = 5), or uninterested in continuing (*n* = 2) (i.e., no specific reason given). Table 1 shows the characteristics of participants. The ranges of the age and BMI of the participants were 44–61 years old and 21–28.5 body mass index (BMI). Participant characteristics reveal no significant differences between the two groups at baseline and between the two groups for those who completed the 1-year follow-up.

Cronbach’s alpha was used to analyze the reliability of the questionnaire. Both the SF-36 (*α* = 0.861) and PA estimates of standard self-report questionnaire (*α* = 0.831) were reliable and valid. Except bodily pain (*α* = 0.613), vitality (*α* = 0.650), and social functioning (*α* = 0.403) dimensions, the remaining SF-36 dimensions, including physical functioning (*α* = 0.828), role-physical (*α* = 0.882), general health (*α* = 0.707), role-emotional (*α* = 0.813), and mental health (*α* = 0.702), had reliability coefficients higher than 0.7. Furthermore, five of the PA estimates of standard self-report questionnaire dimensions, namely, social support (*α* = 0.862), PA hinder (*α* = 0.891), expected accomplishments (*α* = 0.914), PA enjoyment (*α* = 0.800), and self-efficacy (*α* = 0.932), had reliability coefficients higher than 0.8.

### 3.2. Health Outcome

Quality of life (QOL): To examine the differences of QOL between the ME and UC, linear mixed models were performed to verify whether QOL was better in participants with ME. The results showed significant differences between the ME and UC, favoring ME for vitality-related QOL (5.776, 95% CI 0.987−10.565, ES = 0.360) and mental health-related QOL (9.938, CI 4.146, 15.729, ES = 0.512) after the one-year intervention. There were no other significant differences found for both the ME and UC (Table 2).

Muscle strength: the linear mixed models were performed (Table 3) and observed significant differences between the ME and UC, favoring ME for leg strength and endurance (2.880, CI 1.337, 4.423, ES = 0.557) and chair stand test or arm lifting test (ALT) for strength and endurance of upper extremities (2.745, CI 1.076, 4.415, ES = 0.491).

Cardiorespiratory endurance: Table 3 presents the differences of cardiorespiratory endurance between the ME and UC. However, there were no significant differences between the groups for cardiorespiratory capacity.

Physical activity estimates: Table 3 shows the differences of PA estimates between the ME and UC groups. Significant differences were found in both the ME and UC, favoring ME for PA hindrance (5.120, 95% CI 1.976, 8.264, ES = 0.486). Other significant differences were not found between the ME and UC.

## 4. Discussion

This study focused on the long-term effects of diversified combination exercises with remote guidance (e.g., resistance training, aerobic exercises, and psychological guidance) on breast-cancer patients’ QOL, muscle strength, cardiopulmonary endurance, and participation of PA in China. The results demonstrated that, after one year, the experimental group showed some favorable differences in QOL, muscle strength, cardiopulmonary endurance, and participation of PAs, and especially improvement of physical capability and mental health QOL, significantly raised muscle strength of the upper and lower limbs and reduced obstruction in PAs, but no significant difference between groups in cardiopulmonary endurance was observed.

According to the analysis of QOL, some indices between groups were of no statistical significance, but the experimental group had a more significant improvement in quality of life than the control group. It is worth noting that quality of life for the experimental group was markedly enhanced based on energy and mental health statistics compared with the UC group. Previous studies have indicated that the main cause of fatigue in breast cancer patients is due to surgery and chemotherapy during treatment [26]. Fatigue is almost universal among the breast cancer patients [27,28], which continues long after treatment, even after 10 years of treatment [29,30,31]. The persistent fatigue not only severely interferes with patients’ normal physical function [32] but also has a significant correlation with the occurrence of depression and anxiety in later stages [33], which greatly affects the QOL of breast cancer patients [34]. Therefore, the improvement of physical energy and endurance are critical for breast cancer patients because undergoing surgery or chemotherapy during treatment may cause many mental health problems which heavily influence daily life [35,36]. The results of the study illustrated that combination exercises with remote guidance for breast cancer patients yields a certain improvement in the QOL after one year. In addition, the functional limitations caused by physiological problems were also reduced in the experimental group.

Regarding strength, the significant differences in strength of the upper and lower limbs between the experimental and the control groups remained and the participants in the experimental group improved greatly. As far as we know, there have been fewer studies exploring the longitudinal effects of exercise interventions on strength for breast cancer patients. This study investigated the longitudinal effect of internet-based combination exercise programs on muscle strength among breast cancer patients after one year. The results showed that combination exercises with remote guidance have long-term and significant effects on fatigue and strength of the upper and lower limbs of breast cancer patients after surgery. Findings of this study are consistent with previous studies in that general fatigue is associated with muscle strength in breast cancer patients [37] and that muscle strength is a predictor of death for older adults [38]. In addition, some studies have found that muscle atrophy can cause physical impairment [39,40], disability [41,42], fall risk [43], loss of independence [44], and decreased QOL [45] among older people and is a major determinant of muscle atrophy. During the one-year follow-up, lymphedema was not reported among participants in the experimental group, which is consistent with previous studies showing that resistance training does not result in lymphedema—a condition common in breast cancer patients [46,47,48]. We speculate that the difference may be due to the improvement of muscle strength and QOL in the experimental group so that they can gradually take care of themselves and even return to work normally. Then, they may become conscious of the benefits of exercise and may therefore adhere to exercise actively for a long time such that physical capability may be rehabilitated significantly after one year.

Cardiopulmonary function is an important predictor of cardiovascular events and mortality [49,50]. The treatment of breast cancer leads to decreases in cardiopulmonary function [51], but regular PA can offset such a decrease and even improve cardiopulmonary functions [52,53,54]. However, after the one year of follow-up, improvement of patients in the experimental group was higher than in the control group, but statistical analysis demonstrated that there was no significant difference between two groups and that the effect size was small. We speculate that, during the follow-up, the experimental group may have engaged in more PA than the control group, but there may have been nuances of exercise regularity and intensity between groups, which warrants further study in the future.

Physical inactivity is a predictor of death both for breast cancer patients and healthy women [49,50]. Some studies have demonstrated that PA can improve physical fitness and QOL in women [55,56,57,58,59], with clinical significance for breast cancer patients [60]. In the present study, the experimental group participating in PA had a certain improvement over the control group with obstruction factors of PA observing a significant difference between groups. The experimental group showed that PA barriers were reduced greatly and that the level of PA increased appropriately. Previous studies have indicated that PA can improve health, symptoms control, QOL, and complications after treatment among breast cancer patients [55,56].

Our findings suggest that taking part in planned remote exercise guidance intervention after breast cancer surgery may provide long-term benefits for some physical and psychological conditions. Both rehabilitation professionals and breast cancer patients should be acutely aware of the benefits of combined exercises [61]. Considering the lack of physical activity in breast cancer patients, it is necessary to formulate corresponding exercise prescriptions and behavioral intervention strategies [62,63] during the whole recovery process to guarantee or improve the level of physical activity.

### Strengths and Limitation

First, it is one of the only longitudinal studies to date that has evaluated the long-term beneficial effects of remotely guided combination exercises (including anti-resistance training, aerobic exercises, and psychological guidance) on postsurgical breast cancer patients. From this perspective, the study is significant because it provides a reference for the implementation of postoperative rehabilitation programs for breast cancer patients in the future. Second, at the one-year follow-up, although about one-fifth of the participants dropped out, the dropout rate was lower than the maximum rate recommended by long-term follow-up studies (30%) [64]. The limitation is that the follow-up time of this study was only one year and thus observation time needs to be extended for further verification.

## 5. Conclusions

After one year, the experimental group improved significantly in energy, mental health, PA, and muscle strength of the upper and lower limbs, but cardiopulmonary endurance was not significantly different between the two groups. The present study suggests that combination exercises with remote guidance leads to long-term enhancement of the QOL and muscle strength among breast cancer patients. A longitudinal study is needed to further explore the long-term impact of remotely guided multicomponent exercise intervention on breast cancer patients after surgery.

## Figures and Tables

**Figure 1 ijerph-17-03425-f001:**
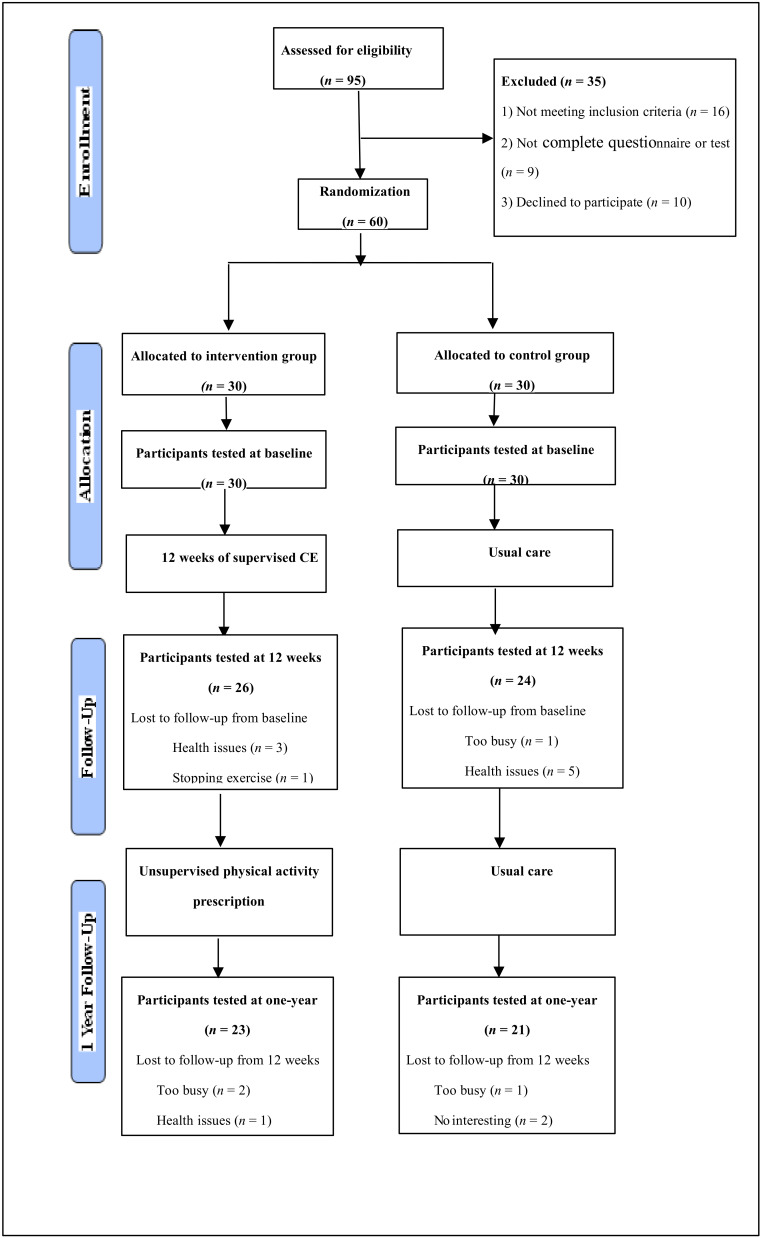
Consolidated standards of reporting trials flowchart of the 2 arms of the study.

**Table 1 ijerph-17-03425-t001:** Participants’ characteristics at baseline and differences between groups.

Variable	All Participants Tested at Baseline			Completers 1-Year Post-Intervention		
	Total (*n* = 44)	ME (*n* = 23)	UC (*n* = 21)	Total (*n* = 44)	ME (*n* = 23)	UC (*n* = 21)
**Anthropometric**						
Age (years)	51.2 (7.9)	50.4 (7.4)	52.0 (8.5)	50.7 (7.0)	48.8 (5.7)	52.8 (7.8)
Weight (kg)	62.9 (7.39)	61.6 (6.32)	64.3 (8.21)	62.9 (7.87)	61.3 (6.53)	64.7 (6.95)
Height (m)	1.6 (0.05)	1.6 (0.06)	1.6 (0.06)	1.6 (0.04)	1.6 (0.05)	1.6 (0.04)
BMI	24.57 (2.60)	24.17 (2.07)	24.98 (3.02)	24.55 (2.48)	23.89 (2.01)	25.27 (2.77)
**Marital Status**						
Married	37/44	19 (82.6%)	18 (85.7%)	39/44	20 (87.0%)	19 (90.5%)
Single	7/44	4 (17.4%)	3 (14.3 %)	5/44	3 (13.0%)	2 (9.5%)
**Blood Pressure**						
SBP at rest (mmHg)	121.3 (20.6)	117.7 (15.8)	125.0 (24.3)	121.6 (17.2)	118.3 (13.9)	125.3 (19.9)
DBP at rest (mmHg)	75.4 (10.6)	77.0 (11.0)	73.9 (10.2)	76.1 (10.0)	77.0 (11.2)	75.2 (8.6)
Mean Blood Pressure	90.7 (12.6)	90.5 (11.6)	90.9 (13.7)	91.3 (10.8)	90.7 (11.0)	91.9 (10.9)
**Stage of illness**						
I	18/60	8 (26.7%)	10 (33.3%)	16/44	6 (26.1%)	10 (47.6%)
Ⅱ	33/60	19 (63.3%)	14 (46.7%)	23/44	15 (65.2%)	8 (38.1%)
Ⅲ	9/60	3 (10.0%)	6 (20.0%)	5/44	2 (8.7%)	3 (14.3%)
**PT**						
Observation	7/44	3 (13.0%)	4 (19.0%)	5/44	2 (8.7%)	3 (14.3%)
Chemotherapy	15/44	7 (30.4%)	8 (38.1%)	18/44	9 (39.1%)	9 (42.9%)
Radiation therapy	1/44	1 (4.3%)	0(0%)	0/44	0 (0%)	0 (0%)
Radio-chemotherapy	21/44	12 (52.2%)	9 (42.9%)	20/44	12 (52.2%)	8 (38.1%)

SD, standard deviation; ME, multicomponent exercise group; UC, usual care group; BMI, Body Mass Index; SBP, systolic blood pressure; DBP, diastolic blood pressure; PT, Phase of Treatment; KG, Kilogram. Values are *n*, mean (SD), or as otherwise indicated. Statistical evaluations were made by Analysis of Variance.

**Table 2 ijerph-17-03425-t002:** Quality of life 1-year post-baseline.

Variables	Group	Baseline Mean (SD)	1 Year Mean (SD)	Baseline to 1 Year		
				Between-Group Differences Mean Change (95% CI)	ES	*p*-Value
SF-36						
PF	ME	82.61 (11.27)	90.00 (8.12)	3.219 (−2.969, 9.408)	0.155	0.305
	UC	78.81 (20.24)	79.76 (22.83)			
RP	ME	29.35 (36.66)	67.39 (32.36)	16.046 (−0.570, 32.661)	0.288	0.058
	UC	58.33 (44.25)	64.29 (42.26)			
BP	ME	72.17 (13.80)	80.43 (11.07)	−0.631 (−6.932, 5.669)	−0.030	0.843
	UC	72.38 (18.95)	81.90 (14.70)			
GH	ME	65.61 (16.71)	75.61 (19.06)	3.786 (−4.287, 11.859)	0.140	0.355
	UC	56.38 (19.36)	58.81 (23.68)			
VT	ME	61.96 (12.41)	80.65 (15.69)	5.776 (0.987, 10.565)	0.360	0.018 *
	UC	63.81 (7.05)	70.95 (13.47)			
SF	ME	88.04 (17.87)	105.98 (22.57)	1.527 (−7.750, 10.804)	0.049	0.745
	UC	84.52 (25.28)	99.40 (26.95)			
RE	ME	63.77 (41.33)	79.71 (35.87)	3.996 (−12.661, 20.654)	0.072	0.636
	UC	55.56 (43.89)	63.33 (40.32)			
MH	ME	51.48 (6.19)	78.78 (16.90)	9.938 (4.146, 15.729)	0.512	0.001 **
	UC	52.76 (6.53)	60.19 (25.06)			
HT	ME	3.52 (1.04)	4.13 (0.87)	0.281 (−0.141, 0.702)	0.198	0.191
	UC	3.43 (1.25)	3.48 (0.87)			

SD, standard deviation; CI, confidence interval; ME, multicomponent exercise group; UC, usual care group; SF-36, the Mos 36-item Short Form Health Survey; PF, Physical Functioning; RP, Role-Physical; BP, Bodily Pain; GH, General Health; VT, Vitality; SF, Social Functioning; RE, Role-Emotional; MH, Mental Health; HT, Reported Health Transition; ES effect size; * *p* < 0.05 and ** *p* < 0.01.

**Table 3 ijerph-17-03425-t003:** Physical Activity (PA) estimates, muscle strength, and cardiopulmonary endurance 1-year post-baseline.

Variables	Group	Baseline Mean (SD)	1 Year Mean (SD)	Baseline to 1 Year		
				Between-Group Differences Mean Change (95% CI)	ES	*p*-Value
SPSDCT	ME	14.96 (2.96)	22.48 (4.87)	2.880 (1.337, 4.423)	0.557	0.000 **
	UC	15.57 (2.66)	17.33 (3.31)			
ALT	ME	15.91 (4.99)	22.26 (3.85)	2.745 (1.076, 4.415)	0.491	0.001 **
	UC	19.05 (3.29)	19.90 (3.43)			
VO2max	ME	41.82 (18.89)	51.72 (16.91)	2.500 (−4.213, 9.213)	0.111	0.462
	UC	42.78 (18.04)	47.76 (14.08)			
PA Estimates						
Social support	ME	15.83 (5.33)	18.04 (4.41)	0.228 (−2.036, 2.492)	0.030	0.843
	UC	13.38 (6.79)	15.14 (5.93)			
PA hinder	ME	27.17 (4.75)	45.65 (6.04)	5.120 (1.976, 8.264)	0.486	0.002 **
	UC	28.57 (5.27)	36.81 (7.45)			
EA	ME	34.83 (2.19)	36.65 (3.98)	0.461 (−1.188, 2.110)	0.083	0.581
	UC	34.57 (4.95)	35.48 (2.96)			
PA enjoyment	ME	17.39 (2.02)	18.43 (2.69)	0.093 (−0.903, 1.089)	0.028	0.853
	UC	17.76 (3.02)	18.62 (1.56)			
Self-efficacy	ME	70.52 (19.65)	71.52 (14.14)	−1.524(−10.502, 7.454)	−0.051	0.738
	UC	57.67 (25.38)	61.71 (24.72)			

SD, standard deviation; CI, confidence intervals; ME, multicomponent exercise group; UC, usual care group; EA, expected accomplishments; VO2, max maximal oxygen uptake; SPSDCT, stand-up and sit-down chair test (number of times standing up from the chair within 30 s); ALT, arm lifting test (30 s dumbbell of 5 pounds or 2.3 kg lifting test); ES, effect size; ** *p* < 0.01.

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
