# Peer review of "A Longitudinal Study of a Multicomponent Exercise Intervention with Remote Guidance among Breast Cancer Patients"

_ijerph, 2020, doi:10.3390/ijerph17103425_

Round 1
Reviewer 1 Report
The authors have examined the impact of both aerobic and resistance exercise training with Remote Guidance on psycho-physiological factors for health care in breast cancer patients. This study is not only a very interesting story and benefit but also a good design. This reviewer has some comments as below.
<Major comments>
- In the title, "combined exercise" is a vague expression for readers. Please rephrase it.
- This reviewer understands that the combination of aerobic and resistance exercise is a beneficial strategy for human health care, but it can not be read in this manuscript. Please explain why combined exercise would be good compared with only aerobic or resistance exercise.
- How important is the 12 weeks data? If it is not important, please consider removing 12 weeks of data from Tables and add a supplemental table for 12 weeks of data.
- Pre data is n = 30, while Post data is n = 23 or 21. However, the authors have performed repeated t-test, and thus, the authors have probably used n = 23 or 21 in Pre data in terms of statistical analysis. If so, please indicate n = 23 or 21 data for Pre data, even though the authors collected the n = 30 data.
- Please explain about effect size in statistical analysis. (e.g., Cohen's d)
<Minor comments>
- How important is married status?
- Please add BMI data.
- "Hemodynamics" is a little bit strange because the authors measured only blood pressure. In addition, mean blood pressure should calculate.
Author Response
Dear Reviewer,
Thank you very much for your comments concerning our manuscript. Those comments are all valuable and very helpful for revising and improving our paper, as well as the important guiding significance to our researches. We have studied comments carefully and have made correction which we hope meet with approval. Revised portion are marked in red in the paper. The main corrections in the paper and the responds to your comments are as flowing:
1. In the title, "combined exercise" is a vague expression for readers. Please rephrase it.
* Answer: we have revised in the manuscript, thank you.
- This reviewer understands that the combination of aerobic and resistance exercise is a beneficial strategy for human health care, but it can not be read in this manuscript. Please explain why combined exercise would be good compared with only aerobic or resistance exercise.
* Answer: we have explained it, please see L56-60.
- How important is the 12 weeks data? If it is not important, please consider removing 12 weeks of data from Tables and add a supplemental table for 12 weeks of data.
* Answer: we have removed 12 weeks of data from Tables and added a supplemental table for 12 weeks of data, please see L275-283.
- Pre data is n = 30, while Post data is n = 23 or 21. However, the authors have performed repeated t-test, and thus, the authors have probably used n = 23 or 21 in Pre data in terms of statistical analysis. If so, please indicate n = 23 or 21 data for Pre data, even though the authors collected the n = 30 data.
* Answer: we have indicated n = 23 or 21 data for Pre data, please see L159 (Table 1).
- Please explain about effect size in statistical analysis. (e.g., Cohen's d)
* Answer: we have explained about effect size in statistical analysis, please see L143.
<Minor comments>
1.How important is married status?
* Answer: because marital status has a great impact on the physical and psychological status of breast cancer patients, this study used it as a baseline characteristics to analyze whether there was any difference between groups when randomizing.
- Please add BMI data.
* Answer: we have added BMI data, please see L159 (Table 1).
- "Hemodynamics" is a little bit strange because the authors measured only blood pressure. In addition, mean blood pressure should calculate.
* Answer: we have revised in the manuscript. The mean systolic blood pressure and mean diastolic blood pressure have been calculated, please see L159 (Table 1).
Reviewer 2 Report
The aim of study is very interesting for the scientific community, but for its publication I consider it necessary to polish and complete some sections that I review below:
In the Materials and Methods section, it is necessary to complete or improve the following aspects.
In Participants:
- The type of sampling carried out must be indicated.
- It remains to be reviewed or clarified if the sample had previous sports physical activity and whether the experimental group in addition to the intervention program, carried out another type of exercise or sports physical activity.
In section 2.2 (Designed intervention program), the intervention program of the control group in terms of practice time, intensities, heart rate, among other parameters that allow the reader to know in more detail the weekly training carried out, needs to be further detailed.
In section 2. 4 (Outcome measures): Justify why SF-36 has been used to measure the quality of life of women with breast cancer and not FACT-B, which is a specific questionnaire to assess quality of life in this population (women with breast cancer).
In the Results section, it is necessary to indicate the reliability (omega) of each of the factors of the questionnaires used in this study.
Author Response
Dear Reviewer,
Thank you very much for your comments concerning our manuscript. Those comments are all valuable and very helpful for revising and improving our paper, as well as the important guiding significance to our researches. We have studied comments carefully and have made correction which we hope meet with approval. Revised portion are marked in red in the paper. The main corrections in the paper and the responds to your comments are as flowing:
In Participants:
- The type of sampling carried out must be indicated.
*Answer: We have revised in the manuscript. Please see P.2, line 82.
- It remains to be reviewed or clarified if the sample had previous sports physical activity and whether the experimental group in addition to the intervention program, carried out another type of exercise or sports physical activity.
*Answer: We have revised in the manuscript. Experimental group do not participate in special physical activity training program except for the intervention program. Please see P.2, line 89; P.3, line 109-110.
- In section 2.2 (Designed intervention program), the intervention program of the control group in terms of practice time, intensities, heart rate, among other parameters that allow the reader to know in more detail the weekly training carried out, needs to be further detailed.
*Answer: We have revised in the manuscript, please see P.3, line 100-107.
- In section 2. 4 (Outcome measures): Justify why SF-36 has been used to measure the quality of life of women with breast cancer and not FACT-B, which is a specific questionnaire to assess quality of life in this population (women with breast cancer).
*Answer: Based on previous studies, we think that SF-36 or FACT-B can be used to evaluate the quality of life of breast cancer [1, 2]. In this study, it is also appropriate to choose SF-36 to evaluate the quality of life, and previous study used FACT-B to evaluate the functional status, while SF-36 to evaluate the quality of life [3]. Thank you.
[1] Fenlon, Deborah , et al. "The JACS prospective cohort study of newly diagnosed women with breast cancer investigating joint and muscle pain, aches, and stiffness: pain and quality of life after primary surgery and before adjuvant treatment." Bmc Cancer 14.1(2014).
[2] Navneet, Kaur , et al. "Survivorship issues as determinants of quality of life after breast cancer treatment: Report from a limited resource setting." Breast 41(2018):120-126.
[3] Argenbright, Christine A , R. E. Taylor-Piliae , and L. J. Loescher . "Bowenwork for symptom management of women breast cancer survivors with lymphedema: A pilot study." Complementary therapies in clinical practice 25(2016):142-149.
- In the Results section, it is necessary to indicate the reliability (omega) of each of the factors of the questionnaires used in this study.
*Answer: The questionnaire of this study has been tested by reliability and validity, and has been applied to similar research. In outcome measures section, we have explained and quoted relevant references, please see P.3, line 129; P.3, line 133.

Reviewer 3 Report
This publications describe the results of a RCT including a remote guidance PA intervention among breast cancer patient. The paper is well written and clear, but from my point of view too straightforward and centered solely on effectivness of the program. The description of the intervention and of the design has to be found in previous publications. The introduction is short, with a high amount of shortcut and the writting could be improved. The method is clear, but refering to other publication, limiting the understanding of the content of the interventions. The results are solely focusing on pre- and post- comparison of the two groups, without taking cofounders or specific components of the intervention into account. The discussion is short, including key references.
Specific comments
L46: extremely imperative is probably a bit too much for a scientific paper. Reference could be added here.
L48: What is authors definition of holistic treatment, what does it entails?
L48-L62: the flow of idea is not clear here, I would start with the benefits of PA among cancer patient and its specificity, then move forward on the effect of social media and remote video to increase PA, then on long term effect. But as such, this is very difficult to follow. Moreover, how PA can help in decreasing the long term psychological and physiological issues and complication after operation is not described in the present paper, only stated. I would add these points to support the understanding and innovation in your paper.
L58-L60; is there any reasons for previous study not using remote guidance? What are the advantages? Which are the limits? Why do you think it is important and effective?
L64: what is meant by a long term comparison?
L64: why comparing with usual care and not with a general PA intervention? How can we distinguish the effect of the components of the intervention? How far did the authors analysed the mechanisms and their interactions in this intervention?
L102: Not sure doctor is the appropriate term?
L138: Maybe the sample size needs to be justified here, as the number of recruited, as well as the final sample are rather low. I would add in the sample description more about power analysis and sampling.
Table 1: While the table describe the sample, no statistical analysis was made to identify differences between CE and UC group, which could explain part of the results. Could these tests be added in the table. Moreover, if significant differences are found, they need to be added to the data analysis comparing the evolution of the two groups, as covariables.
Table 2 and table 3: Why did the authors made t-test and not multivariate analysis? Was any confounder added, variables? Why did the authors not compared between 12 weeks and 1 years (having three times points)? Where there some effects on subsample (stage of cancer)?
L232: What are the mechanisms and key lessons learnt from this articles? Where there some interactions with the medical staff? Were some specific components that worked better for specific population?
Author Response
Dear Reviewer,
Thank you very much for your comments concerning our manuscript. Those comments are all valuable and very helpful for revising and improving our paper, as well as the important guiding significance to our researches. We have studied comments carefully and have made correction which we hope meet with approval. Revised portion are marked in red in the paper. The main corrections in the paper and the responds to your comments are as flowing:
- *L46: extremely imperative is probably a bit too much for a scientific paper. Reference could be added here.
*Answer: we have deleted “extremely” and added reference in the manuscript, please see P.2, line 46-47.
- *L48: What is authors definition of holistic treatment, what does it entails?
*Answer: we have revised “holistic treatment” to “appropriate treatment” in the manuscript, please see P.2, line 47.
- *L48-L62: the flow of idea is not clear here, I would start with the benefits of PA among cancer patient and its specificity, then move forward on the effect of social media and remote video to increase PA, then on long term effect. But as such, this is very difficult to follow. Moreover, how PA can help in decreasing the long term psychological and physiological issues and complication after operation is not described in the present paper, only stated. I would add these points to support the understanding and innovation in your paper.
*Answer: We have revised in the manuscript, please see P.2, line 55-70.
- *L58-L60; is there any reasons for previous study not using remote guidance? What are the advantages? Which are the limits? Why do you think it is important and effective?
*Answer: I have explained it, please see P.2, line 61-65.
- *L64: what is meant by a long-term comparison?
*Answer: we have revised “a long term comparison” to “a long-term follow-up” in the manuscript, please see P.2, line 71.
- 6. *L64: why comparing with usual care and not with a general PA intervention? How can we distinguish the effect of the components of the intervention? How far did the authors analyze the mechanisms and their interactions in this intervention?
*Answer: Thank you for this question. In this study, because we did not restrict their voluntary movement \general physical activity of UC group, we did not compare with a general PA intervention in the UC group. We use this way to distinguish the effect of the components of the intervention.
We have analyzed the mechanisms and their interactions in this intervention, please see P.9, line 230-232; P.10, line 244-246; P.10, line 250-251.
- *L102: Not sure doctor is the appropriate term?
*Answer: We have changed “a doctor” into “treatment requirement ” in the manuscript, please see P.3, line 114.
- *L138: Maybe the sample size needs to be justified here, as the number of recruited, as well as the final sample are rather low. I would add in the sample description more about power analysis and sampling.
*Answer: we have added in the sample description more about power analysis and sampling, please see P.3, line 135-138.
- *Table 1: While the table describe the sample, no statistical analysis was made to identify differences between CE and UC group, which could explain part of the results. Could these tests be added in the table. Moreover, if significant differences are found, they need to be added to the data analysis comparing the evolution of the two groups, as covariables.
*Answer: We have added “Table 1 shows the characteristics of participants. The ranges of the age and BMI of the participants were 44–61 years old and 21–28.5 BMI, respectively. Participant characteristics reveals no significant differences between the two groups at baseline, and between the two groups for those who completed the 1-year follow-up.” in the manuscript, please see P.4, line 153-156.
- *Table 2 and table 3: Why did the authors made t-test and not multivariate analysis? Was any confounder added, variables? Why did the authors not compared between 12 weeks and 1 years (having three times points)? Where there some effects on subsample (stage of cancer)?
*Answer: (1) We used the mixed linear effect model to compare the differences between the two groups, please see line 140-141.
(2) Any confounder was not added because treatment variables and potential outcome is independent.
(3) The aim of this study was to determine whether there were any differences in muscle strength, cardiorespiratory endurance, QoL and physical activity (PA) barriers between the multicomponent exercise and usual care after 1 year post-baseline. because we have analyzed its benefits in our previous article regarding after 12 weeks of intervention, we did not repeat it again. In addition, we have removed 12 weeks of data from Tables and add a supplemental table for 12 weeks of data, please see P.10, line 275-283.
We had previously considered the stages of cancer, only to assess whether there was a difference between the two groups at random, so this study did not consider effects on subsample (stage of cancer). Thank you.
11.*L232: What are the mechanisms and key lessons learnt from this article? Where there some interactions with the medical staff? Were some specific components that worked better for specific population?
*Answer: we have revised in the manuscript. Please see P.10, line 253-258.

Round 2
Reviewer 1 Report
This manuscript is improved after revising. This reviewer has only one minor comment as below.
Please calculate the mean blood pressure. The authors have measured systolic and diastolic blood pressure, so mean blood pressure should be calculated. Formula is [(Systolic - Diastolic)/3 + Diastolic]
Author Response
Dear Reviewer,
Thank you very much for your comments concerning our manuscript. Those comments are all valuable and very helpful for revising and improving our paper, as well as the important guiding significance to our researches. We have studied comments carefully and have made correction which we hope meet with approval. Revised portion are marked in red in the paper. The main corrections in the paper and the responds to your comments are as flowing:
Please calculate the mean blood pressure. The authors have measured systolic and diastolic blood pressure, so mean blood pressure should be calculated. Formula is [(Systolic - Diastolic)/3 + Diastolic].
* Answer: we have calculated the mean blood pressure in the manuscript, please see P.6, line 167 (Table 1). Thank you.
Reviewer 2 Report
The authors have improved the manuscript presented but still do not respond to a fundamental aspect that I asked them, which refers to the reliability of each of the factors of the questionnaires used in the present study. In other words, the reliability (cronbach's alpha and omega) found in their study is missing.
In line 129-133, the reliability found by other researchers in their study is indicated in one of the questionnaires used. For publication, it is necessary to indicate the reliability of each of the factors of all the questionnaires used in the present investigation with your sample.
Author Response
Dear Reviewer,
Thank you very much for your comments concerning our manuscript. Those comments are all valuable and very helpful for revising and improving our paper, as well as the important guiding significance to our researches. We have studied comments carefully and have made correction which we hope meet with approval. Revised portion are marked in red in the paper. The main corrections in the paper and the responds to your comments are as flowing:
*In line 129-133, the reliability found by other researchers in their study is indicated in one of the questionnaires used. For publication, it is necessary to indicate the reliability of each of the factors of all the questionnaires used in the present investigation with your sample.
*Answer: We have indicated the reliability of each of the factors of all the questionnaires used in the present investigation with my sample in the manuscript. Please see P.4, line 157-164.
Cronbach’s alpha was used to analyze the reliability of the questionnaire. Both the SF-36 (α=0.861) and PA estimates standard self-report questionnaire (α=0.831) were reliable and valid. However, the Bodily Pain (α=0.613), Vitality (α=0.650) and Social Functioning (α=0.403) dimensions reliability of SF-36 are a little low. We think the reasons are as follows: 1) The sample size of this study is relatively small; 2) There are only two questions in the dimension of Bodily Pain or Social Functioning. Than you.
Reviewer 3 Report
Thank you to the authors for having answered to my comments.
Author Response
Dear Reviewer,
Thank you very much for your comments concerning our manuscript.